# Numerical Simulation and Experimental Study on Axial Stiffness and Stress Deformation of the Braided Corrugated Hose

**Dacheng Huang and Jianrun Zhang \***

School of Mechanical Engineering, Southeast University, Nanjing 211189, China; huangdacheng22@foxmail.com
\* Correspondence: zhangjr@seu.edu.cn; Tel.: +86-025-5209-0520

**Abstract:** To explore the mechanical properties of the braided corrugated hose, the space curve parametric equation of the braided tube is deduced, specific to the structural features of the braided tube. On this basis, the equivalent braided tube model is proposed based on the same axial stiffness in order to improve the calculational efficiency. The geometric model and the Finite Element Model of the DN25 braided corrugated hose is established. The numerical simulation results are analyzed, and the distribution of the equivalent stress and frictional stress is discussed. The maximum equivalent stress of the braided corrugated hose occurs at the braided tube, with the value of 903MPa. The maximum equivalent stress of the bellows occurs at the area in contact with the braided tube, with the value of 314MPa. The maximum frictional stress between the bellows and the braided tube is 88.46MPa. The tensile experiment of the DN25 braided corrugated hose is performed. The simulation results are in good agreement with test data, with a maximum error of 9.4%, verifying the rationality of the model. The study is helpful to the research of the axial stiffness of the braided corrugated hose and provides the base for wear and life studies on the braided corrugated hose.

**Keywords:** axial stiffness; numerical simulation; equivalent model; braided corrugated hose; metallic braided tube

## 1. Introduction

Braided corrugated hose is used in many places as an important mechanical component, such as chemical plants, the automobile industry and the aerospace industry [1,2]. Due to the special structure of the braided corrugated hose, it can minimize vibration and compensate displacement when suffering from the dynamic load caused by gas and liquid.

The braided corrugated hose is mainly composed of three parts: metal bellows, the metallic braided tube, and joint. Figure 1 shows the schematic diagram of the braided corrugated hose. Metal bellows are made using a forming process applied on thin metal sheets and it can compensate for the displacement when suffering axial loads. The metallic braided tube is braided with wires on the braiding machine, and it can protect the internal bellows, minimize the vibration, and withstand the axial loads. The bellows and the braided tube are connected by joints at both ends.

According to the literature, scholars have thoroughly studied the modeling of the bellows. They have mainly used the mechanical analysis modeling and finite element methods to study the mechanical properties of the bellows. Hachemi [3] studied the effect of plastic strain and residual stress after hydroforming on cyclic life fatigue. He found that as the elongation increases, the number of cycles decreases, and the numerical estimation is lower than the experimental results and the localization of the crack weak area can be detected. Yuan [4] proposed a mathematic description of the reinforced S-shaped bellows and discussed the mechanical characteristics under internal pressure, forced axial, and bending displacement. He found that the pressure capability of the bellows decreases slightly with the increase of the number of layers, but the decrease is

limited and the S-shaped bellows tends to have weak nonlinear characteristics under axial tensile. Hao [5] studied the failure life under the repeating of the bending process. He found that the failure point of the corrugated metal hose was located at the wave trough during the ultimate repeated bending process, and the cracks were distributed along the circumferential direction. The crack propagation and the failure mechanism of the hose under bending deformation was characterized.

Although the previous scholars have done a lot of research on the mechanical properties and fatigue characteristics of the bellows, the use of the metallic braided tube leads to a major difference in the mechanical properties of the braided corrugated hose and the bellows. Nowadays, the simulation studies on the metallic braided tube can be divided into two categories, one of which treats metallic braided tubes as a homogeneous composite material. The constitutive model of the braided tube was established by tensile test. Hachemi [3] treated the braid as a discrete diamond cell unit by finding the relationship between braiding angle and axial strain in theoretical derivation, and solved the constitutive model of the braid using the UMAT subroutine in ABAQUS. It was found that the constitutive model is in good agreement with the test results when the elongation is less than 30 mm. Based on the constitutive model of Hachemi, Hu [6] proposed the acceleration coefficient to modify the change of the braiding angle and explained the mechanical mechanism of inconsistent experimental and simulation results through lateral contact between metal fibers. Other methods for simulation studies of the braided tube were to create the quasi-entity braided tube model in finite element software. The quasi-entity model of the braid can be divided into two types, one method is to set up the finite model with each wire in the braid, and the other is to set up the equivalent model with each strand in the braid. Rial [7] established three models of the metal braid, the micro-scale model, the mesoscale model, and the macro-scale model, respectively. The micro-scale model was generated by strands where each wire was considered as independent structure, the meso-scale model was generated by strands where each group of wires was modelled as a continuum material with equivalent mechanical behavior, and the macro-scale model was generated by homogeneous material. Two experimental tests were established to show the tension and the pressure expansion behaviors, and the finite element approaches were compared with the experiment. The tensile test results showed that the macro-scale model was better able to exhibit small displacement in tension, and the micro-scale and the meso-scale models can show the complex behaviors of wires in the braided tube. Zhao [8] established the braided tube Finite Element Model based on APDL(NASYS Parametric Design Language), and studied the mutual effects between the weaving net, bellows, and armed ring under pressure, as well as the enhancement mechanism. The theoretical analysis method based on the FEM results was proposed to analyze the enhancement capacity quantitatively and check the intensity of the weaving net, which established the relationship between the radial displacement, hoop stress, equivalent pressure, and frictional resistance. Results showed that the weaving net can share 20% of internal pressure 10 MPa, and the weaving net was in the state of elastic deformation. Stephen [9] analyzed the cause of premature failure of the braided corrugated hose under circling pressure and combined the physical evidence with finite element analysis to find that bellows were more likely to burst prior to the braid if there existed too many of gaps or too little of coefficient of friction.

The previous scholars seldom studied the braided corrugated hose, due to the large amount of contacts between the wires and the bellows, which will cause non-convergence of the finite element analysis. Moreover, there is no theoretical basis for generating a simplified model of the braided tube, which can lead to inaccurate calculations. Therefore, it is important to explore a method to generate and simplify the braided tube, which can analyze the axial stiffness and the stress distribution of the braided corrugated hose.

In this paper, the equation of the spatial curve of the braided tube is derived, and the model of the braided tube is established according to the equation spatial curve. The simplified equivalent braided tube model is carried out according to the axial deformation of the wire in the braided tube, and then the Finite Element Model of braided corrugated

hose is established to analyze the axial stiffness and distribution of stress. Then, the tensile experiment of the braided corrugated hose is performed to verify the model, which provides a basis for the wear and fatigue calculations on the braided corrugated hose.

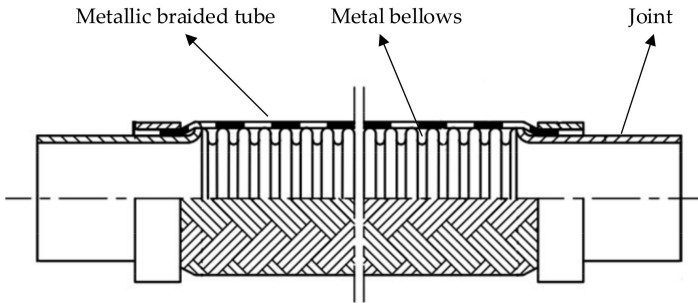

**Figure 1.** Schematic diagram of the braided corrugated hose.

## 2. Material and Methods Description

### 2.1. Material

The material used in the braided corrugated hose is 304 stainless steel in ASTM/AISI. The chemical composition is shown in Table 1. The elastic modulus E = 196 GPa, and Poisson's ration $\mu = 0.3$.

**Table 1.** The chemical composition of 304 stainless steel.

| Specification | C | Si | Mn | P | S | Cr | Ni |
|---|---|---|---|---|---|---|---|
| 304 stainless steel | $\leq 0.08$ | $\leq 1$ | $\leq 2$ | $\leq 0.045$ | $\leq 0.03$ | $\leq 18 - 20$ | $\leq 8 - 10.5$ |

### 2.2. Methods Description

The equivalent method of the braided tube based on the same axial stiffness was proposed. The Finite Element theory was used to simulate the axial stiffness and stress deformation of the braided corrugated hose. The axial tensile test of the braided corrugated hose was carried out to validate the numerical results.

## 3. Modeling of the Metallic Braided Tube

### 3.1. Mathematical Model of the Metallic Braided Tube

The structure of the braided corrugated hose is shown in Figure 1; the metallic braided hose covers the outside of the bellows to protect the internal structure and there is a contact between the metallic braided hose and the internal bellows. The metallic braided hose is made up of several strands of metal wires woven in a certain order and angle along the axis of the bellows. Depending on the desired mechanical properties of the braided corrugated hoses, the geometry, size, the weaving angle, and the number of wires per strand vary greatly, which makes it more difficult to generate geometric models. Therefore, a metallic braided modeling method based on the modified helix line is proposed.

Figure 2 shows the metallic braided tube's basic component. Each strand is constituted by wires, and the wires in each strand have the same motion trajectory. By analyzing the structure of the metallic braided tube, this crossed spiral structure can be equated to the trajectory formed by any point on the outer circumference of the metallic braided corrugated hose rotating around the central axis and moving along the central line at the same time, so the shape of any wire in the braided tube is a cylindrical spiral. Therefore, the wire's motion trajectory can be described in Figure 3, in which *r* is the radius of the

braided tube, $p$ is the pitch, $\alpha$ is the braiding angle, and $\theta$ is the angle of rotation. The equation of motion [10] can be expressed as:

$$\begin{cases} x = r\cos(\theta) \\ y = r\sin(\theta) \\ z = \frac{\theta}{2\pi}\cdot p \end{cases} \tag{1}$$

For the braided tube, the braiding angle depends on the value of pitch $p$. When $p$ is large, the weaving speed is fast, so the braiding angle is small; when $p$ is small, the weaving speed is slow, so the weaving angle is large. The relationship between braiding angle $\alpha$ and pitch $p$ can be expressed as:

$$\tan(\alpha) = \frac{2\pi r}{p} \tag{2}$$

Formula (1) is the wire trajectory in one direction, while the metallic braided tube is interlaced in two directions. Therefore, by transforming the sign in the y-direction of Formula (1), the other direction of the wire trajectory can be obtained. For the places where the clockwise and counterclockwise wires intersect, the previous scholar [11] used the constraint equation to constrain the degree where they coincide, but this method did not quite fit compared with the realistic situation. According to some research [12], a sinusoidal motion can be added to the helix line. Indeed, the radius of the helix line is not constant, and it will become bigger or smaller like a sine function. To avoid the clockwise and counterclockwise wires intersecting at the same point, we set different direction's wires to have different radii, as Figure 4 shows; therefore, the helix lines in different directions will not interference when intersecting, and the improved radius Formula of helix line can be expressed as:

$$r = r_0 + h\cdot\sin(N\cdot\theta) \tag{3}$$

where $r_0$ is the nominal radius, $h$ is the maximum distance to the nominal radius, and $N$ is the number of jumps accomplished by the fibers in one pitch length. The complete geometric equation of modified helix line can be expressed as:

$$\begin{cases} x = (r_0 + h\cdot\sin(N\cdot\theta))\cdot\cos(\theta) \\ y = (r_0 + h\cdot\sin(N\cdot\theta))\cdot\sin(\theta) \\ z = \frac{\theta}{2\pi}\cdot p \end{cases} \tag{4}$$

According to Formula (4), the modified helix line is shown in Figure 4, in which the red line is the modified helix and the blue line is the unmodified helix; $h$ is the maximum value of the difference in radii between modified and unmodified helix line, and it guarantees that the different wire can cross without interference. The whole braided tube model can be obtained in SolidWorks, as Figure 4 shows.

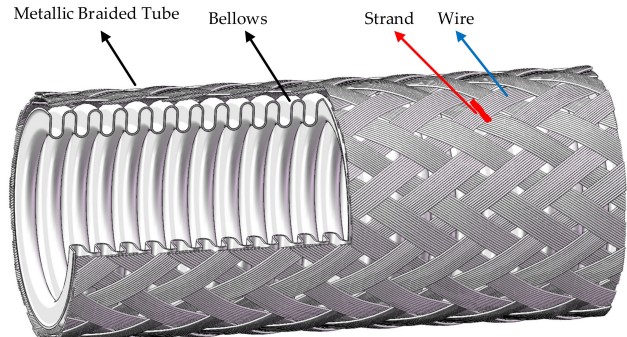

**Figure 2.** The braided corrugated hose.

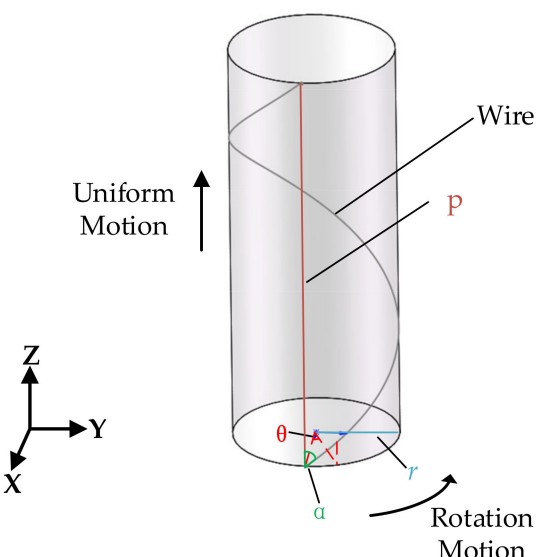

**Figure 3.** The trajectory of the braiding wire.

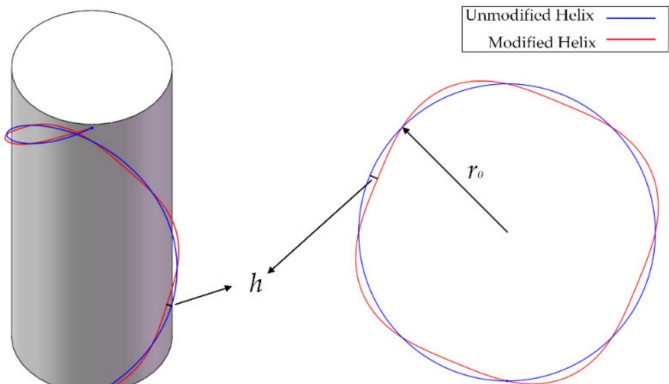

**Figure 4.** Modified helix line.

*3.2. Equivalent Theory Based on Axial Deformation*

The key to generating the equivalent strand model is to obtain the axial deformation of the braided tubes. Therefore, it assumes one strand as one wire for force analysis and the section shape of the wire is still a circle. Figure 5a shows the helix wire, where *r* is the radius of rotation. For the sake of analysis, the unfolded helix wire is used, as Figure 5b shows, where OD is the length of the projection of the wire on the XY plane, which can be expressed as $2\pi r$; DA is the pitch, which can be expressed as *P*; OA is the length of wire, which can be expressed in term of *S*; and $\alpha$ is the initial braiding angle. When the metallic braided tube is stretched by axial force, $F_{axis}$ is the force along the axial direction, $F_{wire}$ is the force along the metallic wire's direction, and the relationship between *S*, *P*, *r*, and $\alpha$ based on the triangle can be expressed as:

$$S = 2\pi r \cdot \sin\alpha \tag{5}$$

$$P = \frac{2\pi r}{\tan\alpha} \tag{6}$$

$$F_{axis} = F_{wire} \cdot \cos\alpha \tag{7}$$

When the metallic braided tube is subjected to axial force, the braiding angle changes from $\alpha$ to $\alpha_0$, the axial length changes from *P* to $P_0$, and the length of the wire changes from

*S* to $S_0$. Since both ends of the metallic braided tube are welded to the joint, the projection length of the wire on the XY plane remains unchanged.

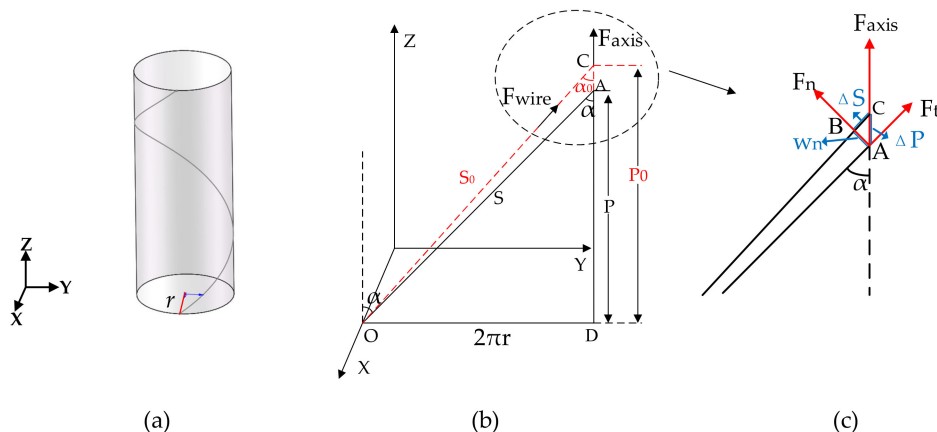

(a)                  (b)                  (c)

**Figure 5.** Force analysis of one wire: (**a**) Diagram of helix wire, (**b**) Diagram of unfolded wire, (**c**) Decomposition of force.

Figure 5c shows the detailed force analysis of the wire. Supposing that the wire OA is fixed at O, the force $F_{axis}$ acts on the point A in the direction of AC, and the initial braiding angle is $\alpha$. In the initial stage, $F_{axis}$ can be decomposed into two separate forces $F_n$ and $F_t$ along different directions, $F_n$ along the normal direction and $F_t$ along the wire direction. Therefore, the axial displacement $\Delta P$ is a combination of displacement in two directions, $w_n$ caused by normal force $F_n$, and the other is $\Delta S$, which is caused by wire force $F_t$, where $w_n$ and $\Delta S$ can be obtained by deflection calculation of a cantilever beam loaded with a concentrated force according to the laws of material mechanics [13].

$$w_n = \frac{F_n S^3}{3EI} \tag{8}$$

where $S$ is the length of wire, $E$ is the elastic modulus, and $I$ is the cross-sectional moment of inertia. When the cross section is a solid circular section, the Moment of Inertia is $I = \frac{\pi d^4}{64}$, where $d$ is the diameter of the section.

$$\Delta S = S \cdot \varepsilon_t = \frac{F_t S}{EA} \tag{9}$$

$$\varepsilon_t = \frac{F_t}{EA} \tag{10}$$

where $A$ is the cross-sectional area, $\Delta S$ is the axial elongation, and $\varepsilon_t$ is the strain along the wire.

According to the Pythagorean theorem, the elongation of the braided tube $\Delta P$ can be expressed as:

$$\Delta P = \sqrt{w_n{}^2 + \Delta S^2} \tag{11}$$

$\Delta P$ is the axial displacement generated by the wire under axial force. Then, the axial stiffness curve of the metallic braided tube can be obtained by comparing F and $\Delta P$.

The above derived axial displacement is based on the equivalence of treating one strand as a single wire for force analysis, while in the actual metallic braided tube, each wire in the strand is suffering axial force. For force analysis of multiple wires, it is assumed that each wire in the strand is independent and will not be disturbed by other wires when suffering axial force. Figure 6 shows the axial force diagram, in which the blue lines are the wires' deformations after suffering axial force. When multiple wires are suffering axial force $F_{axis}$, each wire in the strand will move the same distance $\Delta P'$, respectively, and the braiding angle will change from $\alpha$ to $\alpha\prime$, with each of those is parallel to each other,

so the axial force is uniformly distributed on each wire. Assuming there are $m$ metallic wires in one strand, such that the force on each wire is $\frac{F_{axis}}{m}$, then according to the previous Formulas (8), (9), and (11), displacement in different direction can be obtained:

$$w_{n'} = \frac{F_n S^3}{3mEI'} \tag{12}$$

$$\Delta S' = S \cdot \varepsilon_{t'} = \frac{F_t S}{mEA'} \tag{13}$$

$$\Delta P' = \sqrt{w_{n'}{}^2 + \Delta S'^2} \tag{14}$$

where $I' = \frac{\pi d'^4}{64}$, $A' = \pi \cdot \left(\frac{d'}{2}\right)^2$, and $d'$ is the diameter of one wire per strand.

The axial displacement model of one wire and multiple wires have already been obtained, respectively. To generate the strand equivalent model substituting multiple wires, the axial displacement of equivalent model and multiple wires model should be the same. Keeping the section of one strand equivalent model circular, the elongation of one strand model and multiple wires model should be the same, so:

$$\Delta P = \Delta P' \tag{15}$$

$$d = \sqrt{\frac{3md'^4\sin\alpha \pm \sqrt{\left(3md'^4\sin\alpha\right)^2 + 4\left(16S^2\cos\alpha + 3d'^2\sin\alpha\right)\left(16mS^2\cos\alpha d'^4\right)}}{2\left(16S^2\cos\alpha + 3d'^2\sin\alpha\right)}} \tag{16}$$

By solving Formula (15), the relationship between the diameter $d$ of one strand equivalent model and the diameter $d'$ of the multiple wires model can be expressed as Formula (16).

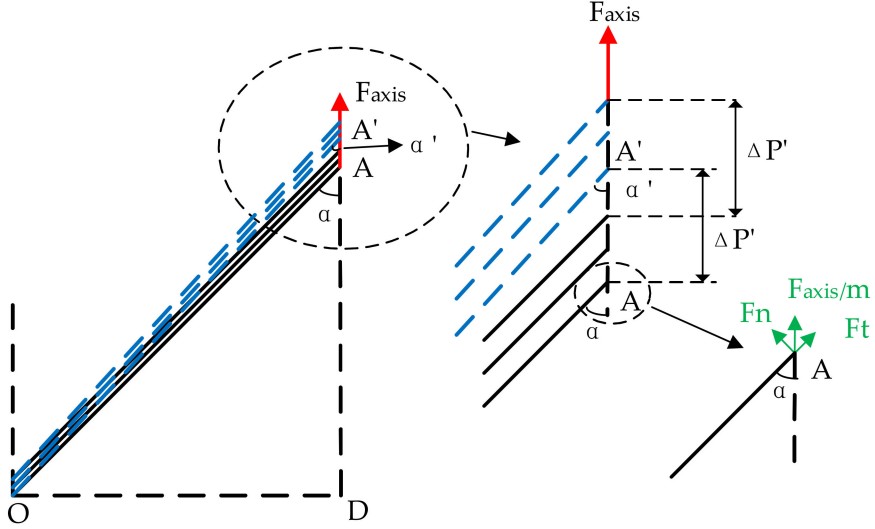

**Figure 6.** Force analysis of multiple wires.

### 3.3. Geometric Model of the Equivalent Braided Corrugated Hose

The DN25 braided corrugated hose consists of two parts: the bellows have an inner diameter of 25.4 mm and the outer diameter of 31.5 mm. The structure and dimensions of the bellows are shown in Figure 7 and Table 2. The braided tube consists of 36 strands, with 12 wires in each strand, and the diameter of each wire is 0.3 mm. The nominal diameter of the braided tube is 33 mm, and the pitch is 80 mm. According to Equation (16), an equivalent braided tube model can be obtained with one wire in each strand, and the equivalent diameter is 0.56 mm.

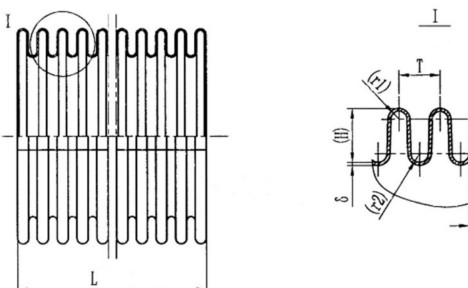

**Figure 7.** The structure of the DN25 bellows.

**Table 2.** The dimensions of the DN25 bellows.

|  | d /mm | D /mm | T /mm | t /mm | r1 /mm | r2 /mm | $\delta$ /mm | H /mm | L /mm |
|---|---|---|---|---|---|---|---|---|---|
| DN25 | 25.4 | 31.5 | 3.7 | 2.1 | 0.8 | 0.8 | 0.25 | 2.8 | 200 |

According to the deduced space curve equation expressing the braided tube, the three-dimensional model of the braided corrugated hose can be built in SolidWorks as shown in Figure 8.

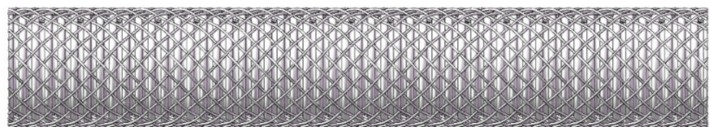

**Figure 8.** Geometry model of DN25 braided corrugated hose.

## 4. Numerical Simulation Analysis and Results

### 4.1. Finite Element Model of the Metallic Braided Corrugated Hose

The material used in the braided corrugated hose is 304 stainless steel in ASTM/AISI. The thickness of the bellows is 0.25 mm, and the total length of the bellows is 200 mm. The thickness of the bellows is much less than the length, and a shell element can be applied when the effect of thickness is not taken into account in numerical simulation. Therefore, the shell element is adopted for the bellows and the 0.25 mm mesh is applied to ensure the quality of the mesh. In order to create face-to-face contact, the solid element is applied to mesh the braided wires and the size is 0.5 mm. The total mesh elements are 288,596 and the nodes are 710,596.

The mesh element simulates the curved shapes of the braided corrugated hose with good agreement. The contact type between the braided wires and the bellows is frictional with a frictional coefficient of 0.2 and a face-to-face contact detection method. The contact type between the braided wires is bonded to limit the large relative movement between wire and wire in a different strand. In element analysis software, the fixed joint was used at the end of the braided tube and the bellows to ensure the braided tube and the bellows can move simultaneously. One end face has constrained freedom in x, y, and z axes, and the other end face has constrained freedom in x and y axes. In this way, this end face can only move in z-direction. A 5 mm axial displacement is applied on this end face. Thus, the Finite Element Model of the braided corrugated hose can be constrained, as shown in Figure 9.

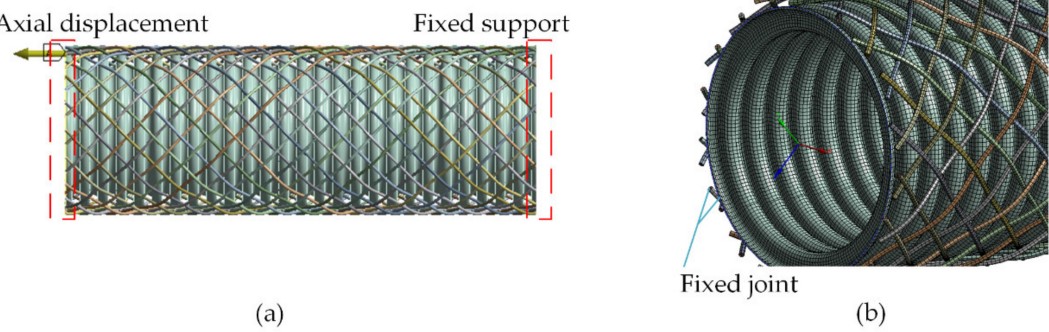

**Figure 9.** Finite Element Model of the DN25 braided corrugated hose: (**a**) Boundary condition, (**b**) Fixed joint.

### 4.2. Axial Stiffness and Stress Distribution Rules

By using the above Finite Element Model for numerical simulation, the equivalent stress and the axial stress of the braided corrugated hose can be obtained, as shown in Figure 10. Figure 11 shows the equivalent stress, and the axial stress of the bellows. Figure 12 shows the axial stiffness of the braided corrugated hose.

The following can be seen from the numerical simulation results:

1.  The equivalent stress of the braided corrugated hose is larger than the bellows under the same axial displacement. This is probably because the outer braided tube is more prone to deformation and carries more stress, as Figures 10a and 11b show. In addition, the equivalent stress changes in the range of 1–903 MPa, and the axial stress changes in the range of −561 to 517 MPa.
2.  For the bellows, the stresses are more likely to be concentrated at the trough of the bellows, except for the area where the crest and the braid come into contact. The stress is unevenly distributed in the wire and is larger when the wire comes into contact with the bellows.
3.  The axial stiffness is a nonlinear curve, as Figure 12 shows, when the stretching distance is within 1.5 mm. The stiffness is linear when the stretching distance exceeds 1.5 mm and increases steeply due to friction contact between the braided tube and the bellows; the inner bellows limit the radial deformation of the braided tube.

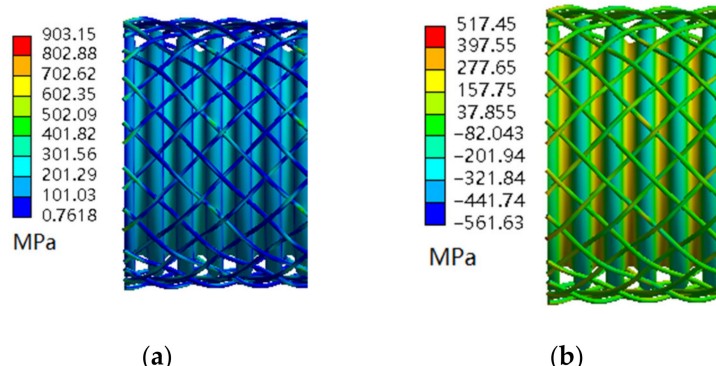

**Figure 10.** Stress diagrams of the DN25 braided corrugated hose: (**a**) Equivalent stress, (**b**) Axial stress.

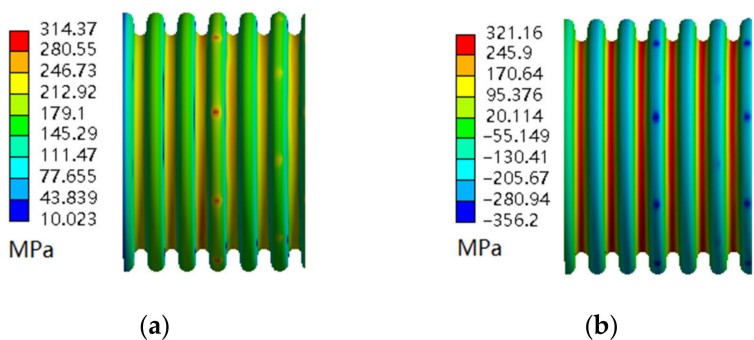

**Figure 11.** Stress diagrams of the DN25 bellows: (**a**) Equivalent stress, (**b**) Axial stress.

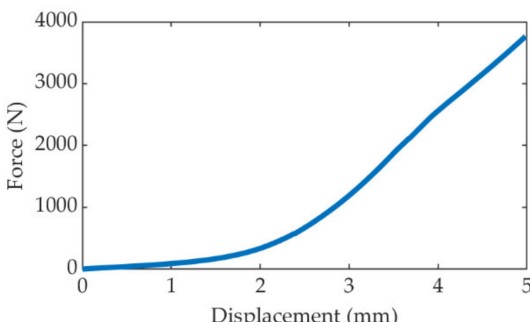

**Figure 12.** Axial stiffness of the DN25 braided corrugated hose.

*4.3. Contact Compressive Stress, Friction Stress, and Distributions*

Considering that the bellows are often worn through by the metallic braided tube in service of use, in order to understand the effect of contact pressure and frictional stress on the bellows, the distributions of contact stress are displayed in Figure 13.

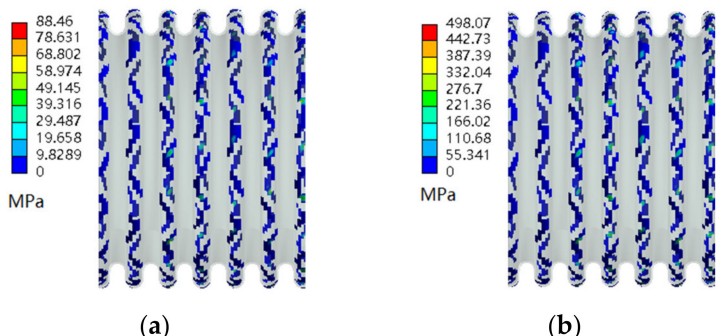

**Figure 13.** Frictional stress and contact pressure of the braided corrugated hose: (**a**) Frictional stress, (**b**) Contact pressure.

The frictional stress, and the contact pressure are mainly distributed at the crest of the bellows, which is the easiest area to make contact with the braided tube. The distribution of the frictional stress is discrete, and the contact position is not the same for each crest, which is mainly determined by the shape of the braided tube.

On the contact surface of the bellows, the frictional stress is distributed like concentric circles. At the center of the contact area, the maximum frictional stress of 88.46 MPa occurs. The frictional stress decreases from the center to the outside of the surface.

In addition, the distribution of the contact pressure is similar to the frictional stress, as Figure 13b demonstrates. The maximum contact stress occurs where the maximum frictional stress occurs, and the maximum value is 498.07 MPa. Therefore, these contact

areas of the bellows are more prone to experience wear and fatigue damage during the service life of the braided corrugated hose.

## 5. Tensile Testing and Result Analysis of the Braided Corrugated Hose

*5.1. Test Scheme*

The ETM-204C computer controlled universal testing machine is used to perform the tensile test on the braided corrugated hose, and it is directly clamped with the tester, as Figure 14 shows. The sample is uniformly stretched (displacement rate is 10 mm/min) and the load-deformation curve of the braided corrugated hose under the axial tensile is obtained.

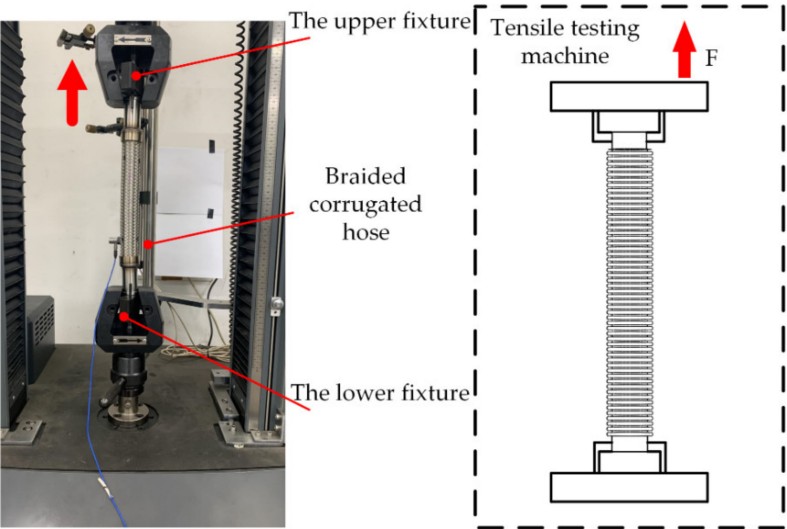

**Figure 14.** Tensile testing apparatus.

The test sample is shown in Figure 15, and the dimensions of the sample are shown in Table 1. Five braided corrugated hoses are stretched, and the average value is calculated.



**Figure 15.** The test sample.

*5.2. Tensile Test Results of the Braided Corrugated Hose under Different Loads*

The braided corrugated hose of 200 mm in length is used to undergo the tensile testing at the displacement of 5 mm. The tensile test results and the simulation results are shown in Figure 16.

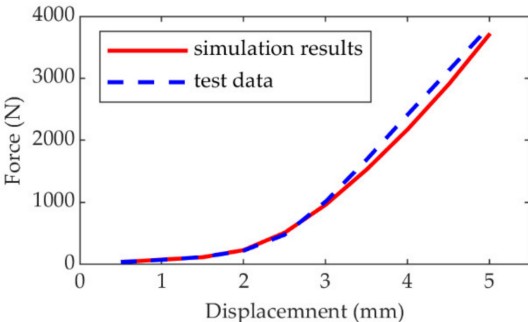

**Figure 16.** Comparation with test results and numerical simulation results.

*5.3. Analysis of Test Results*

The axial stiffness of the braided corrugated hose is nonlinear due to the metallic braided tube. When the stretching length is within 1 mm, the axial stiffness consists of the bellows and the braided tube separately, and the stiffness of the bellows plays a major role. When the stretching length exceeds 2 mm, the friction contact occurs between the bellows and the braided tube, and the bellows also resist the radial deformation of the braided tube. They are main reason for huge stiffness of the braided corrugated hose. Therefore, the axial stiffness of the braided corrugated hose is nonlinear.

The comparison between the test results and the numerical simulation results shows that the trend between the experimental and simulation data is consistent. However, some errors remain, with a maximum error of 9.4% (comparison of the difference in simulation force and test force when stretched to the same displacement. $error = \frac{|F_s - F_e|}{F_e}$, where $F_s$ is the simulation force and $F_e$ is the experimental force). The errors are mainly due to the simplification of the model structure, the equivalence of the braided tube, and the size of mesh generation.

## 6. Conclusions

In this study, numerical simulations and experimental studies on the axial and stress deformation of the braided corrugated hose were carried out and the following results were obtained:

1. By analyzing the structural features of the braided tube, the space curve parametric equation of braided tube was deduced. By analyzing the axial deformation of the braided tube, the equivalent braided tube model was generated.
2. A Finite Element Model of DN25 braided corrugated hose was established, and the axial stiffness and the stress distribution were obtained. The axial stiffness of the braided corrugated hose is nonlinear, and the equivalent stress was mainly distributed at the trough of the bellows. The contact pressure and frictional stress were mainly distributed at the crest of the bellows.
3. The tensile experiment on the braided corrugated hose was performed and compared to the numerical simulation results. The experimental data and simulation results are in good agreement and within the permissible error range, verifying the rationality of the Finite Element Model.
4. This study can provide a reference for modeling the braided corrugated hose, especially for the simplification of the braided tube. It can also provide a basis for the wear and fatigue tests of the braided corrugated hose.

**Author Contributions:** This study was initiated and designed by D.H., the numerical simulations, data analysis, and writing the paper were completed by D.H., J.Z. provided suggestions for research design and paper writing. All authors have read and agreed to the published version of the manuscript.

**Funding:** This research was funded by National Key Research and Development Plan, grant number No. 2019YFB2006402.

**Institutional Review Board Statement:** Not applicable.

**Informed Consent Statement:** Not applicable.

**Data Availability Statement:** The data are part of an ongoing project.

**Conflicts of Interest:** The author(s) declared no potential conflict of interest with respect to the research, authorship, and/or publication of this article. The authors declare that they have no conflict of interest.

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
