# Peer review of "Numerical Simulation and Experimental Study on Axial Stiffness and Stress Deformation of the Braided Corrugated Hose"

_applsci, doi:10.3390/app11104709_

Round 1
Reviewer 1 Report
The paper is very interesting. Only a couple of minor remarks:
- the definition of r, \theta and p are repeated before and after equation (1)
- after equation (1) I expect "where", not "Where"
Reviewer 2 Report
The paper aims to evaluate the mechanical features of the braided corrugated pipe by proposing an equivalent braided tube model within the Finite Element software to improve computational efficiency. Numerical simulations and experimental tests are performed in order to know the distribution of equivalent stresses and stresses due to friction.
In my opinion, this research is certainly of interest to the Journal and the provided outcomes can be useful in evaluating the axial stiffness of the braided corrugated pipe. Moreover, is gives good bases for further studies on the wear resistance and life cycle of the braided tube.
However, the overall organization of the paper should be improved for guaranteeing better reading and better comprehension. Furthermore, excessive repetitions along the whole body of the text should be avoided and the punctuation must be checked (major revision).
In the following, the required revisions are listed:
- Please, correct the word “behavior” with the British English variant “behaviour” along the whole body of the manuscript.
- Please, replace the words “modeling” and “modeled” with the British English variants “modelling” and “modelled” along the whole body of the manuscript.
- Please, use the capital letter for “Finite Element Model” along the whole body of the text.
- Please, revise the text avoiding word repetitions and paying attention to punctuation.
- Please add the definite article “the” before the word “braided” along the whole body of the text.
- Please add the full stop at the end of all the captions.
- Please correct the progressive number of paragraphs 3.1 to 3.3.
- Please centre the figures in relation to the text.
- Introduction, page 1. In the second sentence please remove the definite article before the word “displacement”.
- Introduction, page 1. In the third sentence please replace the comma with a full stop between “joint” and “Figure 1”.
- Introduction, page 2. In the sentence “Metal bellows are…” please replace the comma with the conjunction between “sheets” and “it”, and the comma with the full stop between “loads” and “metallic”. Also, please add the definite article before “metallic braided tube” and replace the comma with the full stop between “loads” and “the”.
- Introduction, page 2. In the sentence “They are mainly used…” please correct “are” with “have” and delete the article before “mechanical analysis”.
- Introduction, page 2. In the sentence “Hao studied…” please add “the” before “repeating” and “of the” before “bending”.
Introduction, page 2. In the sentence “However, the use…” please add “the” before “metallic”.
- Introduction, page 2. In the sentence “Nowadays the simulation studies…” please add the indefinite article “a” before “homogeneous” and add the full stop after “material”.
- Introduction, page 2. In the sentence “Such as Hachemi…” please correct “discreate” with “discrete” and add the definite article before the words “relationship”, “constitutive” and “braid”. In addition, please remove the comma after “cell unit”.
- Introduction, page 2. In the sentence “Based on constitutive model of Hachemi…” please add the definite article between “on” and “constitutive”. In addition please remove the comma after “angle”. In the same sentence, it is suggested to correct the word “fibers” with the British English variant “fibres”.
- Introduction, page 2. In the sentence “Other methods for simulations…” please replace the comma with the full stop between “software” and “the”.
- Introduction, page 2. In the sentence “Rial established three models…” please add the article before “metal braid”.
- Introduction, page 2. In the following sentence please add the definite article before “micro-scale”, “meso-scale” and “macro-scale”. In addition, please complete the sentence by adding “strands where” between “by” and “each wire” and correct the word “where” with “was”.
- Introduction, page 3. In the sentence “The tensile test results…” please add the definite article before “micro-scale”.
- Introduction, page 3. In the sentence “Zhao established…” please explain the meaning of the acronym APDL (ANSYS Parametric Design Language).
- Introduction, page 3. Please correct “weave net” with “weaving net” along the whole body of the paragraph.
- Introduction, page 3. In the sentence “The theoretical analysis method…” please correct “were” with “was”.
- Introduction, page 3. In the sentence “Stephen analyzed the cause…” please replace “much” with “many”.
- Introduction, page 3. In the next sentence please add “of” between “amount” and “contacts”.
- Section 2.1, page 4. In the first sentence please replace “as” with “is” and correct “Figure 2” with “Figure 1”. Also, please replace the comma with the full stop after the reference to the figure number and replace the comma with “and” between “structure” and “there”.
- Section 2.1, page 4. In the sentence “Depending on the desired…” please add the definite article before “number of wires”
- Section 2.1, page 4. In the sentence “Therefore, a metallic…” please add the indefinite article before “modified”.
- Section 2.1, page 5. In the first sentence please replace the comma with the full stop after “component”. In addition, the next part of the sentence “the metallic braided tube is made up of several strands of metal wires woven in a certain order and angle along the axis of the bellows” is quoted in exactly the same way at the beginning of the same paragraph. Please change the structure of the sentence and express the content in a different way, or remove it.
- Section 2.1, page 5. In the sentence “Therefore, the wire’s motion…” please correct the part of the sentence “and r is radius of braided tube” with “in which r is the radius of the braided tube”.
- Section 2.1, page 5. In the next sentence please correct “motion equation” with the form “equation of motion”.
- Section 2.1, page 5. In the sentence after formula (1) please add the definite article before “radius” and before “helix”.
- Section 2.1, page 5. In the following sentence please replace the comma with a full stop after “pitch p”.
- Section 2.1, pages 5-6. In the sentence “Therefore, by transforming…” please add the definite article before “wire trajectory”.
- Section 2.1, page 6. In the next sentence please add the article before “constraint equation” and before “realistic situation”. In addition, please replace “however” with “but”.
- Section 2.1, page 6. In the sentence “According to some…” please replace the comma with a full stop after “helix line”. In addition, please replace “that is” with “indeed”, add the definite article before “helix line” and replace the comma after “constant” with the conjunction “and”.
- Section 2.1, page 6. The sentence “When wires in different…” is not clear. Please review the sentence and modify its structure.
- Section 2.1, page 6. After the formula (3) please add “where” at the beginning of the sentence. Also, please add the definite article before “nominal radius” in both cases in the sentence. In addition, what is the meaning of “number of alternating”? It is recommended to use a clearer or more understandable definition. Also, please replace the comma with the full stop before “N can be obtained”. The part of the sentence from “N ca be obtained...” to the end is unclear. Please review the sentence and modify its structure for better clarity.
- Section 2.1, page 6. In the following sentence please use the capital letter after the full stop.
- Section 2.1, page 6. In the sentence before the formula (4) please remove “And” at the beginning of the sentence.
- Section 2.1, page 6. In the sentence “According to formula (4)…” please add “in which” before “red line”, and add the definite article before “modified”, “blue line” and “unmodified”. Also, please replace the comma between “helix line” and “it” with the conjunction.
- Section 2.1, page 6. The sentence “Generate 3D curve…” is a meaningless sentence. Please review the sentence and modify its structure.
- Section 2.2, page 7. In the second sentence please change the verbal tense by replacing “assuming” with “it assumes” thus making it consistent with the sequence of tenses. In addition, please delete the comma after “force analysis”, add the definite article before “wire” and add the indefinite article before “circle”.
- Section 2.2, page 7. In the sentence “Figure 5(a)…” please add the definite article before “radius of rotation” and replace the comma after “rotation” with a full stop. In addition, where is the radius r indicated in the figure? Also, please replace the full stop after “Figure 5 shown” with a comma. At the end of the sentence, please add the definite article before “initial braiding angle”.
- Section 2.2, page 7. In the sentence “When metallic braided…” please add the article before “metallic” and “triangle” and replace the comma with the conjunction between “direction” and “the”. In addition, please add “is the force” after “Faxis” and “Fwire” and add the colon at the end of the sentence, before the formula.
- Section 2.2, page 8. In the second sentence please add the definite article before “XY plane”.
- Section 2.2, page 8. In the sentence “Therefore, the axial displacement…”, for the sake of clarity and completeness, please correct “where wn and ∆? can be obtained by deflection calculation in mechanics of materials” with “where wn and ∆? can be obtained by deflection calculation of a cantilever beam loaded with a concentrated force according to the laws of material mechanics”
- Section 2.2, page 8. In the sentence following the formula (8) please correct “cross sectional” with “cross-sectional” adding a hyphen. In addition, please replace the comma after “inertia” with a full stop. Moreover, for better understanding, please review the sentence replacing “when cross section is circular, the I = , d is the diameter of circular section” with “when the cross-section is a solid circular section, the Moment of Inertia is I = , where d is the diameter of the section”.
- Section 2.2, page 8. In the sentence following the formula (10) please add the definite article before “cross-sectional” and “strain”.
- Section 2.2, page 8. In the formula (11) please correct “wt” with “wn”.
- Section 2.2, page 9. In the first sentence please add “while” before “in the actual metallic braided tube”.
- Section 2.2, page 9. In the second sentence please delete the comma after “multiple wires” and modify the verbal tense replacing “assuming” with “it is assumed that”.
- Section 2.2, page 9. In the sentence “Figure 6 shows…” please add “in which” before “the blue lines” and replace the comma between “axial force” and “when” with a full stop.
- Section 2.2, page 9. In the sentence “The axial displacement model…” please replace the comma between “respectively” and “to generate” with a full stop.
- Section 2.2, page 10. In the sentence “Keeping the section…” please delete “is” between “model” and “circular”.
- Section 2.2, page 10. In the sentence “By solving the formula…” please add the definite article before “diameter d’”.
- Section 2.2, page 10. Please centre the formula (16) in relation to the text.
- Section 2.3, page 10. In the first sentence please correct “is consisted” with “consists” and “has” with “have”. In addition, please add the indefinite article before “outer diameter”.
- Section 2.3, page 10. In the sentence “The nominal diameter…” please use the capital letter at the beginning of the sentence.
- Section 2.3, page 10. In the sentence “According to the equation16…” please use the capital letter at the beginning of the sentence and delete the article before “equation 16”.
- Section 3.1, page 10. It is suggested to include a general outline of the material under examination, and its main mechanical properties and chemical composition. In addition, please specify that this material and its features are governed by the GB T 1220-92 standard (GB, Chinese National Standards issued by the Standardization Administration of China). Moreover, it is suggested to define, as a reference, its equivalent steel grade also according to the European Standard (Steel grade X5CrNi18-10 – N° 1.4301) and the ASTM/AISI (304 ss).
- Section 3.1, page 10. It is suggested to insert the values of Elastic modulus and Poisson’s ratio in the general overview recommended in the previous point.
- Section 3.1, page 10. Please when you talk about the shell181 and solid186 elements, explain what they are and what their features are used for. In this section, the details of the computational Finite Element Model are not clear. Why using these elements? What the mesh size is? How the mesh size has been set? Is the mesh size the same for both types of elements? These points should be explained. Explain also how the used mesh size was selected (sensitivity analysis?).
- Section 3.1, page 10. In the sentence “In ANSYS Finite Element…” please replace “element” with “elements” and “node” with “nodes” when you define the element number.
- Section 3.1, page 11. Please correct the first sentence as follows: “The contact type between braided wires and the bellows is frictional with a frictional coefficient of 0.2 and a face-to-face contact detection method.”
- Section 3.1, page 11. In the sentence “The contact type between…” please delete the comma after “bonded”.
- Section 3.1, page 11. In the sentence “In this way, this end…” please correct “z direction” with “z-direction” adding a hyphen.
- Section 3.2, page 11. In the first and second sentence please remove the commas after “stress”.
- Section 3.2, page 11. In the sentence “Fig 11 shows…” please replace “Fig 1” with “Figure 1” using the same format of the whole body of the text to define the figures.
- Section 3.2, page 11. In the sentence “In addition, the equivalent stress…” please use the capital letter at the beginning of the sentence.
- Section 3.2, page 11. Please correct the last sentence of point n.2 of the numbered list as follows: “The stress is unevenly distributed in the wire and is larger when the wire comes into contact with the bellows”.
- Section 3.2, pages 11-12. In point n.3 of the numbered list please add the indefinite article before “nonlinear curve” and replace the full stop with the comma after “Figure 11 shows”. Also, please use the capital letter at the beginning of the second sentence, remove the commas after “linear” and “1.5mm” and avoid repeating “the stiffness” replacing it with “and”.
- Section 3.3, page 12. In the first sentence please correct “wore” with “worn”.
- Section 3.3, page 13. In the second sentence please replace “discussed” with “displayed”.
- Section 3.3, page 13. In the sentence “Figure 12 shows…” please replace the comma with the full stop before “the frictional stress” and delete the comma after that. In addition, please add the definite article before “easiest area”.
- Section 3.3, page 13. In the sentence “The distribution of the frictional…” please remove the comma after “discrete”.
- Section 3.3, page 13. In the sentence “At the center of the contact…” please remove the comma after “contact area”.
- Section 3.3, page 13. In the sentence “The maximum contact…” please remove the definite article before “frictional stress” and the comma after “occurs”. Moreover, please revise the sentence avoiding excessive repetitions.
- Section 4.1, page 13. Whereas the experimental campaign plays an important role in this study, it is recommended to add a technical overview with the features of the testing machine. Also, it is suggested to add some photos of the experimental campaign, for better comprehension.
- Section 4.1, page 13. In this section, the details of the experimental campaign are poor. What are the specimens’ features (shape, geometry, dimensions)? How many samples have been tested? How many tests have been done? It is also suggested to insert some photos of the specimens used for the tests.
- Section 4.1, page 13. In the first sentence please replace the comma after “hose” with the conjunction “and”.
- Section 4.1, page 13. In the second sentence please delete the commas after “experiment” and “given”.
- Section 4.2, page 14. In the second sentence please correct the verb “is” with “are”.
- Section 4.3, page 14. In the second sentence please add the definite article before “stretching” and correct “is consisted” with “consists”.
- Section 4.3, page 14. In the sentence “The comparation between…” please correct “comparation” with “comparison”.
- Section 4.3, pages 14-15. In the last sentence please delete the comma after “remain” and replace the definite article before “maximum error” with the indefinite article.
- Conclusions, page 15. In the first point of the numbered list please correct “structure” with “structural” and delete the commas after “tube” in both sentences.
- Conclusions, page 15. In the second point of the numbered list please delete the comma after “nonlinear” in both sentences.
- Conclusions, page 15. In the third point of the numbered list please add “to” before “the numerical simulations” and replace “were” with “are” in the second sentence.
- Conclusions, page 15. In the fourth point of the numbered list please use the capital letter at the beginning of the second sentence and add “tests” after “fatigue”.
- Figure 2. The strand and wire elements are not very clear in the drawings. It is advisable to make an enlargement.
Figure 3. To make the drawing more easily visible it is suggested to indicate the elements more clearly, even with different colours, avoiding overlapping lines and arrows on the drawing.
Figure 4. The arrows used to identify the elements make the drawing very confusing and hard to read. It is suggested to use a legend to indicate the meaning of the blue and red lines.
- Figure 5. To make the graphs more easily visible, the figures should be enlarged. In addition, it is suggested to magnify the diagram and use colours for the figure (b) to make the graph more readable. Use the same font writing as other ones for the symbol “wn”. Why two captions? Insert only the letters (a), (b) and (c) below the respective images. In figure (b), what does the symbol P0 refer to? Moreover, replace the comma with the colon after the general description of the caption, e.g. “Figure 5. Force analysis of one wire: (a) Diagram of helix wire, (b) Diagram of unfolded wire, (c) Decomposition of force.”
- Figure 6. The figures should be enlarged and in a better resolution. The lines of the drawing should be drawn with a less marked stroke. To make the diagram more easily visible it is suggested to change the colours used and not overlap the label of the elements on the chart.
- Figure 7. It is suggested to insert a drawing with the dimensions for both the braided corrugated hose and the numerical model.
- Figure 8. The red dashed rectangles do not make it clear what they indicate and the “displacement” is not clear what it refers to. The different elements composing the model (the fixed joint and the applied displacement with its 5 mm value) should be more clearly indicated in the drawing. Moreover, to make the figure more easily visible it is recommended to change the colour and brighten it.
- Figures 9-10-12. Please replace the comma with the colon after the general description of the caption.
- Figures 11-14. In the labels please add spacing between the variable and the unit of measure.
- Figure 13. It is suggested to magnify the image of the testing machine for better comprehension.
- Figure 14. It is recommended to insert the legend next to the graph and not in the caption.
Reviewer 3 Report
Dear Authors,
I carefully read your manuscript. In general, your paper is interesting, especially from the application point of view, but before publication, it has to be significantly improved.
Below I listed all things which in my opinion should be corrected:
- In your abstract, you put only information about your activities that you made during your research. Please put some data about the general state of the art in your topic, then shorten your text about conducted research, and finally include some data about final results.
- Put your keywords in order from general to specific.
- Please review your text regarding English editing, there are a lot of issues, i.e. word braided appear in one sentence a few times. Additionally, sentences are very complex which makes your manuscript difficult to read.
- In the introduction, you put some literature review. There is some short, general information about each research paper. Provide more data - especially, what was concluded or describe milestones that were reached. This issue appears mainly in the first part about hoses - in the part connected with numerical models, it looks fine.
- You put the following information: the properties of this material are stipulated in the national standards of China - provide exact data about this standard. In general, it is better to put ISO standards.
- You highlighted a stress range as 561-517MPa. Make it properly - from lower to a higher value.
- Put units in figures 9,10 and 12.
- Divide your manuscripts into the proper order: Introduction-> Materials and methods description-> theoretical part-> simulations-> validation->conclusions.
- Put data about the tensile testing machine, standards for conducted tests, number of tested parts, data about extensometer or methodology for strain determination.
- If you provide data from numerical analysis in MPa, prepare your chart after tensile testing as megapascals in a function of strain.
- Describe properly a chart in figure 14.
- Please put information about calculations of error in a level of 18.5%.
- There is not attached any information about the novelty of your research. Please provide proper text about it.
Round 2
Reviewer 2 Report
Apart from few editorial issues, Authors followed all my queries. The paper can be accepted
Author Response
Thank you for your valuable comment. I have checked my submission again and change some editorial issues. Thank you again for your valuable comment.
Reviewer 3 Report
Dear Authors,
Your manuscript has been improved but not at a sufficient level.
Below you can find my second-stage comments for the most important issues.
- Comment:
Put your keywords in order from general to specific.
- Reply:
Thank you for your valuable comment. I have reordered my keywords from braided corrugated hose, metallic braided tube, numerical simulation, equivalent model, axial stiffness. Thank you again for your valuable comment.
2nd Comment:
It must be the opposite. From axial stiffness and numerical simulation at the beginning to braided corrugated hose and metallic braided tube at the end. From general to specific.
- Comment:
Put units in figures 9,10 and 12.
- Reply:
Thank you for your valuable comment. I have put the units in the figures 9,10 and 12. You can see the change in my resubmission manuscript by using “Track Changes”. Thank you again for your valuable comment.
2nd Comment:
Units are provided selectively, and some are still missing i.e. There are two units instead of four in Figures 10 and 11.
Comment:
Divide your manuscripts into the proper order: Introduction-> Materials and methods description-> theoretical part-> simulations-> validation->conclusions.
- Reply:
Thank you for your valuable comment. Because my paper did not involve the Materials and methods description, so I put my experiment in the “Validation” section. I have added some materials to make the experiment more detailed. Thank you again for your valuable comment.
2nd Comment:
I cannot agree, you described a material -> 304 stainless steel, methods-> numerical theory, tensile testing machine, etc. I will leave it to the editor’s consideration.
Comment:
Put data about the tensile testing machine, standards for conducted tests, number of tested parts, data about extensometer or methodology for strain determination.
- Reply:
We gratefully appreciate for your comment. I have put some data about the tensile testing machine and experiment in my paper and you can see the changes in my resubmission manuscript by using “Track Changes”. The extra information is the tensile testing machine is made by Chinese manufacturers called “Wance” and the model is ETM-204C. The maximum tensile the tensile tester can provide is 20 KN and the displacement resolution is 0.06 μm. The large deformation relative error is within . I just fixed the sample’s both side on the tester and set an axial displacement 5 mm, then the upper fixture will stretch the sample to 5 mm at a speed of 10 mm/min and the load-deformation will be obtained. We stretched the sample three times to take the average value. We are sorry to do not use strain gage to measure the strain deformation of the braided corrugated because the dimension of the DN25 is very small and the outer side is braided tube. So, we cannot use any method to stick the strain gage and the stress and strain deformation can only been seen in numerical results. Thank you again for your valuable comment.
2nd Comment:
I am sorry to write it but, in my opinion, you have just proven that part of your research has no sense. Additionally, it has no cover in standards or literature (no standards numbers, no citations). You mentioned you tested a sample three times – in standards, there are at least five tests of a different sample. Also, it is necessary to put some average values and their deviations (i.e., standard deviation). I do not know if I understand it right but based on your reply you tested the same sample three times which is unacceptable.
Comment:
If you provide data from numerical analysis in MPa, prepare your chart after tensile testing as megapascals in a function of strain.
- Reply:
Thank you for your valuable comment. I just using tensile testing machine to test the axial stiffness of the braided corrugated hose in order to validate the simulation results. Because the size of DN25 braided corrugated hose is so small and the existence of the braided tube, it is very difficult to measure the stress in the braided corrugated hose. So, I can only analyze the simulated stress distribution. Thank you again for your valuable comment.
2nd Comment:
As I mentioned in the comment above, in my opinion, validation in such a form has no sense and it has not provided reliable and meaningful (valid) results.
Comment:
Please put information about calculations of error in a level of 18.5%.
- Reply:
Thank you for your valuable comment. I have put the information about how to calculate the errors. And the error is calculated by comparison of the difference in simulation force and test force when stretched to the same displacement. Like , where is the simulation force and is the experimental force. Thank you again for your valuable comment.
2nd Comment:
I cannot see such information in the uploaded version.
Round 3
Reviewer 3 Report
The paper could be accepted.